# A Thermoelectric Energy Harvester Based on Microstructured Quasicrystalline Solar Absorber

**DOI:** 10.3390/mi12040393

**Published:** 2021-04-02

**Authors:** Vinícius Silva Oliveira, Marcelo Miranda Camboim, Cleonilson Protasio de Souza, Bruno Alessandro Silva Guedes de Lima, Orlando Baiocchi, Hee-Seok Kim

**Affiliations:** 1Department of Electrical Engineering, Federal University of Paraíba, João Pessoa, PB 5115, Brazil; marcelo.camboim@cear.ufpb.br (M.M.C.); protasio@cear.ufpb.br (C.P.d.S.); 2Department of Mechanical Engineering, Federal University of Paraíba, João Pessoa, PB 5045, Brazil; bruno.guedes@ct.ufpb.br; 3School of Engineering and Technology, University of Washington Tacoma, Tacoma, WA 98195-2180, USA; baiocchi@uw.edu (O.B.); heeskim@uw.edu (H.-S.K.)

**Keywords:** solar absorber, quasicrystal, energy harvesting, thermoelectricity

## Abstract

As solar radiation is the most plentiful energy source on earth, thermoelectric energy harvesting emerges as an interesting solution for the Internet of Things (IoTs) in outdoor applications, particularly using semiconductor thermoelectric generators (TEGs) to power IoT devices. However, when a TEG is under solar radiation, the temperature gradient through TEG is minor, meaning that the TEG is useless. A method to keep a significant temperature gradient on a TEG is by using a solar absorber on one side for heating and a heat sink on the other side. In this paper, a compact TEG-based energy harvester that features a solar absorber based on a new class of solid matter, the so-called quasicrystal (QC), is presented. In addition, a water-cooled heat sink to improve the temperature gradient on the TEG is also proposed. The harvester is connected to a power management circuit that can provide an output voltage of 3 V and store up to 1.38 J in a supercapacitor per day. An experimental evaluation was carried out to compare the performance of the proposed QC-based harvester with another similar harvester but with a solar absorber based on conventional black paint. As a result, the QC-based harvester achieved 28.6% more efficient energy generation and achieved full charge of a supercapacitor around two hours earlier. At last, a study on how much the harvested energy can supply power to a sensor node for Smart agriculture during a day while considering a trade-off between the maximum number of measurements and the maximum number of transmission per day is presented.

## 1. Introduction

The Internet of Things (IoTs) have been in growing development, and hundreds of billions of IoT devices are expected in the next decade [1]. An unsolved question in this area is how to supply power to an IoT device [2], mainly, for environmental applications. In this context, batteries are the usual solution. However, they require regular maintenance [3] and are not environmentslly friendly [4]. In this scenario, energy harvesting techniques has arisen as a potential alternative solution for powering IoTs [5].

Energy harvesting is defined as the capture of wasted energy from surrounding environmental sources as sunlight, heat, Eletromagnetic waves mechanical waves, mechanical motion, etc. The capture of that energy is performed by a transducer that also converts it into electric energy. After the conversion, that energy is conditioned and stored to power ultra-low-power electronic devices [6,7]. Thermoelectric-based energy harvesters have been an attractive way to convert heat into electricity, and the amount of applications powered by thermoelectric energy harvesting has been increasing in the past decade [4,6,7].

In general, thermoelectric energy harvesters are composed of an array of p- and n-type semiconductor thermocouples connected electrically in series and thermally in parallel, which is sandwiched between two ceramic plates (called hot face and cold face). The TEG generates an output voltage integrated from the temperature gradient on the faces of the TEG, taking advantage of the Seebeck effect [5,8,9].

As the most important thermal energy source is solar radiation, it is reasonable to consider the use of TEGs to generate energy directly from sunlight. However, it is observed that, when a TEG is under the same incident solar radiation, both sides become thermally in equilibrium and, in this scenario, a TEG is useless for harvesting energy [5]. The key challenge in solar thermoelectric energy harvesting is how to maintain an effective temperature difference ΔT ($\Delta T=T_{hot}-T_{cold}$, where $T_{hot}$ is the hot-face temperature and $T_{cold}$ is the cold-face temperature, on the TEG’s faces for a prolonged period [10]. A possible solution is to provide a way to selectively absorb solar radiation using coatings that feature high absorbance in a solar radiation spectrum and low emittance in the thermal radiation band [5].

In recent years, several researchers have developed solar absorbers made of different materials. In this work, a novel solar absorber based on quasicrystal is introduced. Quasicrystals (QCs) are a new class of solid materials that can be positioned between crystals and amorphous materials. In a contract of crystals, a quasicrystal has a different lattice structure with up to thousands of atoms in its unit cell. The quasicrystals feature a long-range ordered structure but do not show three-dimensional translational periodicity as crystals do. These new materials present non-crystallographic rotational symmetry patterns [11].

Quasicrystals are a type of metal alloy, but due to their lattice properties, they do not behave as conventional intermetallic compounds [12] and, as a consequence, they exhibit unexpected properties such as low thermal conductivity, low electric conductivity, high corrosion resistance, low sliding friction and adhesion, and high hardness [13]. Another fundamental characteristic of quasicrystals is that they can have their properties adjusted to match some requirements of the application. All these properties make quasicrystals very attractive for coating. In the literature, studies on quasicrystalline coating application propose the use of nanometric coating obtained by sputtering [14]. Despite nanometric quasicrystalline coating presenting several advantages, it is limited to application on small surfaces due to its high cost. One alternative option is to use micrometric quasicrystalline coating, which is used in this work.

Recently, a variety of applications powered by TEG-based energy harvesters has been investigated by several researchers. In [15], an experimental study was carried out to evaluate the performance of the thermoelectric generator TES1-12704 as a thermal energy harvester for powering wearable smart devices placed on the arms and legs. The conducted experiment measured the harvested energy while users performed daily activities, such as sitting, walking, jogging, and riding a bike. The obtained power output was in the range from 5 to 50 μW with a temperature gradient less than 2.5 °C [15].

In [16], a hybrid energy harvester was presentedto improve upon the network lifetime of IoT healthcare devices. The presented solution made use of both the RF energy harvester and the thermal energy harvester to increase the lifetime of devices. The thermal energy harvested at a temperature gradient of 15 °C was about 0.530 J. In general, it has been found that the network lifetime increased by 24% in the hybrid energy harvesting network compared to a network without any energy harvesting [16].

A novel floating device with multi-source energy harvesting (solar panel and thermoelectric generator) was described in [17]. The thermoelectric component of this energy harvesting system consisted of seven TEGs connected electrically in series, exploiting thermal differences created between the water surface and hot side exposed to sunlight. The floating device harvested sufficient energy to be self-sustaining during sunny days, and the thermoelectric generator provided 0.425 Wh to the system [17].

An ultra-low-power multisensor system powered by a solar thermoelectric energy harvester for environmental measurement purposes was introduced in [18]. The presented thermoelectric energy harvesting system included a novel ultra-low power-management circuit, which drained only 5.5 μA, using only commercial solutions. The proposed energy harvesting system could collect and store up to 7 J on a sunny day and could power the multisensor system for more than 11 days without harvesting any energy.

Another issue regarding thermoelectric energy harvesting is how to improve its efficiency. Kraemer et al. [19] was the pioneering work that presented the development of a high-performance solar thermoelectric generator. The developed solar thermoelectric generators achieved a peak efficiency of 4.6% under 1 kW· m${}^{-2}$. To achieve this, the thermoelectric generator was built based on new nanostructured thermoelectric materials, a spectrally selective solar absorber, and a glass vacuum enclosure that exploits high thermal concentration in a vacuum environment.

In [20], a new solar absorber applied to thermoelectric energy harvesting system was introduced. An organic bucky sponge was the base of the solar absorber as the sponge presented a combination of light-absorbing and heat-insulative properties. [20]. The proposed energy harvester consisted of a floating TEG apparatus covered with the solar-absorber sponge. The harvester was evaluated under emulated sunlight, achieving a maximum open-circuit voltage of around 60 mV while 26 mV was obtained without the sponge. The maximum output power was achieved when the external resistance was about 2 Ω, and the output power was 0.32 mW and 0.05 mW, with and without the solar-absorber sponge, respectively [20]. A disadvantage of that solution is the complexity in synthesizing the solar absorber sponge.

In [21], the effectiveness of a high-temperature spectrally selective absorber and its performance on thermoelectric power generation under a high optical concentration ratio was demonstrated. The proposed solar absorber was attached to a stack of three TEGs connected in series. It was placed under a Fresnel lens that can provide an optical concentration ratio of 62. The solar-to-electrical conversion efficiency was 1.2%, achieved with a temperature of 280 °C at the TEG’s hot side at an input solar flux of 845 W· m${}^{-2}$ under matched load condition [21].

Some of the abovementioned works used solar radiation and thermoelectric generators to obtain electricity. References [19,21] proposed TEG systems combined with optical concentration devices to achieve high temperature to work properly but has the disadvantage of high cost of the optical system used. In [18], an anodized aluminum flat panel, which did not show the characteristics of a solar absorber, was used to heat the hot side of the TEG. In [17], the presented solution used a water-based heat sink but did not use any solar absorber attached to the hot side of the TEG.

In this work, a compact solar-absorber-based thermoelectric energy harvester in extension to [5] is presented. The main novelty of the proposed energy harvester is the use of a solar absorber coated by a micrometric quasicrystalline alloy. Additionally, a water-cooled heat sink was proposed to increase the temperature gradient on the TEG. To test the proposed harvester, a complete experimental apparatus was developed to measure the ambient temperature, the temperature of the absorber, the generated voltage, the regulated voltage, and the stored energy. The developed experimental apparatus allows us to carry out comparative studies. In this way, the proposed solar-absorber-based thermoelectric energy harvester was compared with one using a different and simple type of absorber. The obtained experimental results show the effectiveness of the proposed solution, achieving 57.9 °C under an ambient temperature of 32 °C and 50.8% more efficiency than the other solar absorber.

This paper is organized as follows: Section 1 discusses the main ideas of the proposed thermoelectric energy harvester and compares our proposal with related work. Section 2 provides an overview describing the physical structure of the thermoelectric energy harvester, the quasicrystalline solar absorber, and the power conditioning circuit. Section 3 describes the test for comparison of the proposed QC-based harvester’s performance with that of a harvester using a conventional black paint (BP) as a solar absorber. Section 4 reports the experimental results and discusses the performance of the harvesters. Section 5 describes a potential application to which the QC-based harvester can be applied as a power supply for a sensor node in Smart agriculture. Finally, Section 6 provides a summary of the achievements in this study.

## 2. The Thermoelectric Energy Harvester System

The physical structure of the proposed harvester is composed of a TEG sandwiched by two copper blocks, as shown Figure 1. The copper was chosen due to its high thermal conductivity (400 WK${}^{-1}$ m${}^{-1}$) and low heat storage density (3.4 J cm${}^{-3}$ K${}^{-1}$). The novelty of the proposed harvester, as can be observed in Figure 1, is that the upper copper block is fully coated by a micrometric-thick quasicrystalline (QC) alloy. The QC-coated copper block works as a solar absorber converting solar radiation into stored thermal energy (more detail is given in the next section). This whole block is 5 mm thick, and this value was chosen in order to reduce the thermal resistance to transfer heat through the hot side of the TEG. The uncoated bottom block, which is 25 mm thick, can be immersed into a water tank working as a heat sink from the TEG’s cold side, but any colder surface may work as well. The water-immersion uncoated-copper block prevents the proposed harvester from reaching thermal equilibrium quickly, which is an effect observed when both TEG sides are exposed under the same solar radiation. This solution leads to a consistent ΔT across the TEG, generating an open circuit voltage at the TEG’s terminals.

A commercial TEG (namely, TEC1-12706 module manufactured by Hebei I.T. Co., Ltd. (Shanghai, China) 40 × 40 × 3 mm size, 127 thermocouples, and 06 A max) was used in the experimental tests due to its low cost to present the highest power density among commercial TEGs. The TEC1-12706 features a Seebeck coefficient of about 35 mV/°C near room temperature and on an open-circuit situation.

The thermoelectric conversion efficiency depends on the obtained ΔT on the TEG’s sides. Assuming that the thermoelectric materials’ properties are constant at the TEG’s operating temperature range, the maximum efficiency of a thermoelectric generator can be computed as follows:(1)$$\eta_{TEG}=\frac{T_{hot}-T_{cold}}{T_{hot}}\cdot \frac{\sqrt{1+ZT_{ave}}-1}{\sqrt{1+ZT_{ave}}+\frac{T_{hot}}{T_{cold}}}$$
where $T_{hot}$ and $T_{cold}$ are the temperatures of the hot and cold sides of the TEG, respectively, and $ZT_{ave}$ is the figure-of-merit of the thermoelectric device, where *T* is an average temperature, $T_{ave}=\frac{T_{hot}+T_{cold}}{2}$ [10]. The $ZT_{ave}$ is given by the following:(2)$$ZT_{ave}=\frac{S^{2}}{RK}T_{ave}$$
where *S* is the Seebeck coefficient, *R* is the electrical resistance, and *K* is the thermal conductance of the thermoelectric device [1]. The $\eta_{TEG}$, $ZT_{ave}$, and other thermal and electrical parameters of a TEG module similar to that used in this work were characterized in [22]. From Equation (Equation 1), the TEG efficiency, $\eta_{TEG}$, increases as the temperature gradient, $\Delta T=T_{hot}-T_{cold}$, increases, and a way to achieve this is to use a solar absorber for heating and a good heat sink for cooling even in the same solar radiation, as proposed in this work.

### 2.1. The Quasicrystalline Solar Absorber

Thermal storage systems are key components of concentrating solar power (CPS) plants, converting and concentrating energy from the sun into heat to increase the operating temperature [23]. Solar absorbers are a type of thermal storage system, and an ideal solar absorber must have a unit absorptance in the whole solar radiation spectrum (ultra-violet, visible, and near-infrared light) and zero emittance in the mid-infrared region [24]. The efficiency of a real solar absorber is given by the following:(3)$$\eta_{abs}=\alpha -\epsilon \cdot \frac{\sigma (T_{abs}^{4}-T_{amb}^{4})}{Q_{abs}}$$
where α is the absorptance, ϵ is the emittance, σ (=$5.6696\times 10^{-8}\;{\mathrm{Wm}}^{-2}K^{-4})$ is the Stefan–Boltzmann constant, $T_{abs}$ is the solar absorber temperature, $T_{amb}$ is the ambient temperature, and $Q_{abs}$ is the solar radiation heat flux [19]. From Equation (Equation 3), to maximize the photothermal conversion, the solar absorber should have higher solar absorptance to absorb maximum solar energy and lower thermal emittance at the operational temperature to minimize thermal losses [25]. A good solar absorber surface can also be spectrally characterized by featuring low reflectance (ρ ≈ 0) at wavelengths λ<4μm and by high reflectance (ρ ≈ 1) at λ>4μm.

The spectral absorptance and emittance can be expressed according to Kirchoff’s law for thermal radiation in terms of total reflectance ρ (λ, θ, *T*) for opaque materials, and it is given by the following:(4)α(λ,θ,T)=ϵ(λ,θ,T)=1−ρ(λ,θ,T)

In general, the optical property of a solar absorber can be characterized by its reflectance in the wavelength range from 0.3 to 20 μm. Knowing the reflectance, the solar absorptance α in the near-normal incidence angle θ and at a fixed temperature can be computed as follows [25]:(5)$$\alpha =\frac{\int_{0.3\mu m}^{4\mu m}I_{sun}\left(\lambda \right)\left[1-\rho \left(\lambda ,T\right)\right]d\lambda }{\int_{0.3\mu m}^{4\mu m}I_{sun}\left(\lambda \right)}$$

In Equation (Equation 5), the integration interval is limited from 0.3 μm to 4 μm due to this range including 95% solar radiation and $I_{sun}$ is the incident solar radiation at AM1.5G (global standard spectrum) conditions. In the same way, the thermal emittance, ϵ, at the near-normal incidence angle θ and at a fixed temperature, can also be computed, and it is given by
(6)$$\epsilon =\frac{\int_{2.5\mu m}^{20\mu m}\left[1-\rho \left(\lambda ,T\right)\right]B\left(\lambda ,T\right)d\lambda }{\int_{2.5\mu m}^{20\mu m}B\left(\lambda ,T\right)}$$
where the integration interval is the mid-infrared region and it is limited from 2.5 μm to 20 μm. B(λ,T) is the spectral irradiance of a blackbody curve, and it is given by
(7)$$B\left(\lambda ,T\right)=\frac{c_{1}}{\lambda^{5}\left[e^{\frac{c_{2}}{\lambda \cdot T}-1}\right]}$$
where $c_{1}$ = 3.7405 × 10${}^{8}$ Wμm${}^{4}$m${}^{-2}$ and $c_{2}$ = 1.43879 × 10${}^{8}$ Wμm${}^{4}$m${}^{-2}$, which are the Planck’s first and second radiation constants, respectively.

Quasicrystaline alloys in specific configurations show solar absorber behavior; as an example, in [26], an alternating thin films of dielectric and quasicrystalline I–Al–Cu–Fe that can be optimized for use as a protective and absorbent coating on a photothermal converter, thus more efficiently converting the energy of solar radiation into energy thermal, was introduced. Another example is a stack of Al${}_{2}$O${}_{3}$/AlCuFe/Al${}_{2}$O${}_{3}$ films deposited on copper films that have shown 90% solar absorption, low thermal emissivity, and reasonable chemical thermal stability, with results observed up 400 °C, making these candidate materials suitable for high-temperature applications [14]

In this work, the novel solar absorber was built using a spectrally selective coating of a micrometric AlCuFeB alloy in the quasicrystalline phase. The 100 μm thick coating was deposited on a copper substrate (40 × 40 × 5 mm dimension) by using the High-Velocity-Oxygen-Fuel (HVOF) technique. This coating technique was applied to produce micrometric quasicrystalline alloy and was chosen because of its ease of operation, high efficiency, and mainly its low cost for coating large surfaces. Besides this, the HVOF-obtained micrometric quasicrystalline coating was shown to have high hardness, low sliding friction, high corrosion resistance [27,28], and good thermal properties such as high absorbance and low emittance. These properties turn micrometric quasicrystalline coating into a good option for coating solar thermal absorber [5].

### 2.2. Power Conditioning Circuit

Under small temperature gradient values, a TEG generates a low level of DC voltage, so a step-up converter needs to be used to boost it to a voltage level able to power electronic devices. However, a suitable power conditioning circuit for energy harvesting application is concerned not only with converting the generated voltage into useful voltage but also with storing the surplus energy that is not used immediately. Besides that, to increase efficiency, the power conditioning circuit must include a power management system [29].

To simplify the implementation, the commercial Integrated Circuit LTC3108-1 from Linear Technology was chosen. The LTC3108-1 is an integrated solution that contains all the requirements for a power conditioning circuit for energy harvesting applications. The step-up regulator can provide an output voltage that can be programmed to 2.5 V, 3 V, 3.7 V, or 4.5 V, and it is based on a synchronous boost converter with Armstrong’s oscillator by using an external transformer with a turn ratio of 1:100, as shown in Figure 2. The LTC3108-1 is a commercial step-up regulator-integrated circuit that needs very low input voltage to work properly, 20 mV, with no similar circuit on the market. A typical schematic of the LTC 3108-1 for energy harvesting applications with a TEG is presented in Figure 2. The LTC3108-1 can operate with ultra-low input-voltage greater than 20 mV (or even less if the external transformer turn ratio is changed). The LTC3108-1 shows an efficiency that changes from 40% to 15% when the input voltage varies from 20 to 500 mV [30]. The LTC3108-1 uses an internal charge control circuit to control the charge current of the output capacitor on $V_{OUT}$. The LTC3108-1 starts its operation when $V_{AUX}$ reaches a proper level. If the output voltage $V_{OUT}$ is above the previously selected threshold, the charging current is turned off and the main output voltage, $V_{LDO}$, is regulated. However, if $V_{OUT}$ drops below that level and $V_{AUX}$ is greater than its proper level, the charging current circuit is enabled again and the LTC3108-1 begins to charge the output capacitor on $V_{OUT}$ again [9]. To store the surplus energy, the LTC3108-1’s output $V_{STORE}$ can be connected to a supercapacitor that can be charged up to 5.25 V, as shown in Figure 2. Therefore, when the input energy source generated by the TEG in such a case cannot provide enough energy to charge $V_{OUT}$, the supercapacitor is used to power the load connected on $V_{OUT}$ [9]. The LTC3108-1 also includes a low current Low Dropout Regulator to provide a regulated 2.2 V output for powering low-power processors or other low-power circuits.

## 3. Experimental Setup

An experimental study was carried out to evaluate the performance of the proposed thermoelectric energy harvester based on QC in comparison with the performance of another thermoelectric energy harvester with the same structure but with a solar absorber coated with a conventional black paint. It is important to note that the BP-coated absorber features a high absorptance but does not show low thermal emittance as the QC solar absorber does. The experimental setup for testing is shown in Figure 3.

The evaluated parameters for both energy harvesters (QC-based absorber and BP-based absorber) were the temperature (*T*) on the absorber blocks, the generated voltage ($V_{gen}$) by the harvesters, the regulated voltage ($V_{reg}$) by the LTC3108-1, and the stored voltage ($V_{STORE}$) across a 100 mF supercapacitor. In addition, ambient temperature and relative humidity in the air were measured.

The temperatures of the solar absorber blocks were measured using the digital temperature sensor DS18B20 fabricated by Maxim Integrated. This sensor can provide temperature measurement with 0.0625 °C resolution and accuracy of ±0.5 °C from −10 °C to +85 °C. This low-cost sensor does not require any calibration [31]. For both harvesters, a temperature sensor was fixed on the center of the upper face of the respective solar absorber block.

For ambient temperature measurement, a low-power sensor HDC1080 from Texas Instruments, a fully calibrated humidity and temperature sensor integrated to a 14-bit Analog-to-Digital Converter channel, and an I${}^{2}$C interface into a single IC were used. The sensor features high accuracy, ±0.2 °C for temperature, and ±2% for relative humidity measurements. Its low power consumption in measurement mode minimizes the self-heating effect [32].

All these sensors were connected to an Adafruit Feather M0+ board, a development board based on the microcontroller ARM Cortex-M0+ (ATSAMD21G18, 32 bits, 48 MHz), and a LoRa radio module (RFM95, 915 MHz). The voltage measurements were obtained using the microcontroller’s ADC that features a resolution of 50 μV (16 bits/3.3 V). The ADC readings show an offset error of approximately +35 mV that negatively influences the voltage measurement accuracy. To reduce this, the ADC was passed through a calibration process based on the two-point calibration method using a True RMS Voltmeter Agilent U1242B as the reference instrument, resulting in a reduction of the offset error to 0.8 mV. The measurements were performed every 10 s and transmitted by the LoRa radio.

As shown in Figure 3, the experimental apparatus consisted of a sealed wooden box that contained a rectangular plastic bowl filled with 750 mL of water. The QC-based harvester and the BP-based harvesters were installed at the top of the wooden box in a way that the heat sink blocks were fully immersed in the water and only the solar absorbers and the TEG were out of the box. The solar absorbers were both exposed to solar radiation. The electronic circuitry was housed in a water-proof case next to the apparatus. This was composed of the Adafruit Feather M0+ board, the LTC3108-1 developed kit board, and the LoRa transmitter module).

The experimental apparatus shown in Figure 3 was deployed at the rooftop of a building located in the city of Joao Pessoa, Brazil. The LoRa receiver had the same setup as the LoRa transmitter, and it was connected to a computer. The received data were obtained and visualized in real-time using a developed OCTAVE (OCTAVE is almost fully compatible with Matlab)-based application. An overview of the experiment is shown in Figure 4.

## 4. Results and Discussion

The experimental results were obtained on a sunny day with some cloudy moments. The data were obtained for 11 h on 7 June 2020 from 6h00 to 17h00, covering the whole solar period. The measured values for both harvesters were the temperature values $T_{QC}$, $T_{BP}$, and $T_{Amb}$ (the ambient temperature); the generated TEG voltage values $V_{QC}^{gen}$ and $V_{BP}^{gen}$; the regulated voltage values $V_{QC}^{reg}$ and $V_{BP}^{reg}$; and the stored voltage values $V_{QC}^{store}$ and $V_{BP}^{store}$.

The measured temperatures data are shown in Figure 5, and the obtained results confirm the expected behavior of a solar absorber since both solar absorbers achieved a temperature much higher than the ambient temperature. It is interesting to highlight that the solar absorbers, under high temperature, heated the water-cooled heat sink to a temperature higher than the ambient temperature. As the maximum ambient temperature was 32 °C around noon, the temperature of the BP solar absorber achieved 49.2 °C max, with a daily average temperature of 7.57 °C higher than the ambient temperature. On the other hand, the proposed QC solar absorber achieved 57.9 °C max, with a daily average temperature of 11.42 °C higher than the ambient temperature. The proposed QC solar absorber turned out to be 50.8% more efficient than the BP solar absorber.

In Figure 5, it is possible to observe that the temperature of the QC-based solar absorber begins rising before the temperature of the BP-based solar absorber. As can be seen in Equation (Equation 4), the absorbance of a material depends on the solar radiation incident angle to the absorber surface. Therefore, in the morning, the solar radiation incident angle ranges from a minimal up to about normal angle (it depends on its proximity to the Equator). In this period, the absorbance of both materials varies differently, causing them to be heated differently as well, and as a result, the generated voltages are different, as seen in Figure 6.

From Figure 6, it can be observed that the generated voltages by both solar harvesters have the LTC3108-1 as the coupled load. The generated voltage by the BP-based harvester achieved a maximum of V ≈ 49 mV. On the other hand, the generated voltage proposed by the QC-based energy harvester achieved a maximum of V ≈ 63 mV, which means a voltage level 28.6% higher.

As an important result, the electricity generation of the QC-based energy harvester was useful (over 20 mV, which is the minimum limit for the LTC3108-1 to work) for more than 7 h, from 7h15 to 14h30 with a few moments of interruptions due to the random variation in environmental factors such as passing clouds. On the other hand, the generated voltage by a BP-based energy harvester was useful for around 6 hours from 8h10 to 14h00, with the same variations. For both energy harvesters, the step-up regulator of the LTC3108-1 supplied a $V_{OUT}$ of 3 V and stored the surplus energy in the supercapacitor during most of the test, as shown in Figure 7 and Figure 8. The charge of the supercapacitor maintained the LTC3108-1 $V_{OUT}$ at 3 V, and its power supplied the LTC3108-1 internal control circuit.

As shown in Figure 8, the supercapacitor of the QC-based harvester was fully charged faster in comparison to the one of the BP-based harvesters. Additionally, that supercapacitor could maintain its full charge for more than five hours until the TEG voltage lowered to below the minimum voltage of the LTC3108-1. As can be observed in Figure 6 and Figure 8, when $V_{gen}$ was lower than 20 mV, the supercapacitor began to discharge yet maintained a working circuit. Additionally, the BP-based energy harvesting system shows a similar result but with a much lower time.

As most of the harvested energy is stored in the supercapacitors, the following explanations consider only the energy stored by them. Both harvesters achieved a full charge on their respective supercapacitors before noon, as can be observed in Figure 9. The value of the supercapacitor was 100 mF and, in this day time, an energy of 1.38 J was achieved. It is important to notice that the mean energy during the day was 1.13 J for the harvested energy by the QC-based harvester and 0.78 J for the BP-based harvester.

A possible drawback of the proposed harvester can be its weight because it is composed of metal blocks. However, as it must be exposed to the sun radiation and outdoors, this is not a relevant issue since the harvester is supposed to be deployed on the building top or the ground.

## 5. A Proof-of-Concept to Validate the Proposed QC-Based Energy Harvester

Regarding Smart agriculture, two issues are considered crucial to maximize productivity, namely, soil moisture and precipitation [33]. In this way, a sensor node for monitoring weather conditions (temperature, pressure, and air humidity), soil moisture, and rain detection was developed, as shown in Figure 10. This sensor node was developed as a proof-of-concept to validate the feasibility of the proposed QC-based energy harvester as its power supply.

As, in general, the RF transceiver is the main part of a sensor node related to power consumption, the LoRa modulation was chosen due to its long range (in the order of kilometers) and its very low power consumption but with reduced data rate transmission. LoRa modulation features three main parameters: the Spreading Factor (SF), Bandwidth (BW), and Coding Rate (CR), which can be defined at the design stage according to the need for data rate, range, and power consumption [34]. In order to reduce the power consumption, the chosen parameter values were SF (Spreading Factor) = 7, CR (Coding Rate) = 4/5, BW (Bandwidth) = 500 kHz, and Transmit Power = 13 dBm. These defined configuration values provide the highest data rate and the shortest transmission time (time-on-air) and, hence, the least possible energy consumption [34].

To simplify the design process, the development board Adafruit Feather M0+ was chosen since its low-power microcontroller provides a real-time clock (RTC) that is useful in correlating the measurement data with time. The microcontroller drains 4 mA in active mode and only 3 μA in standby mode [35]. The LoRa radio is based on the RFM95 module and drains 29 mA during a single transmission and only 200 nA in sleep mode [36].

The ultra-low-power BME680 module manufactured by Bosch was used to obtain the weather condition measured data. The BME680 features fast response and high accuracy and is a digital 4-in-1 sensor measuring gas, humidity, pressure, and temperature [37]. For low-power applications, the BME680 can be set to run in a forced mode where the temperature, pressure, and humidity measurements are performed sequentially, and it automatically returns to sleep mode afterward. In this mode, the average current consumption is only 4 μA for 20 ms and, in sleep mode, is 150 nA [37]. For monitoring soil moisture and precipitation, two low-cost sensors were used, namely, the capacitive soil moisture sensor SEN0193 and a simple resistive rain sensor. The outputs of each sensor were read by the ADC of the Feather M0+ board at a rate of 350 kSps. These sensors consumed 10 mA during each measurement, but the ADC conversion time was only about 100 μs.

In order to estimate how many measurements and transmissions can be performed by the sensor node in 24 h, taking in consideration the total energy provided by the proposed harvester in this time period, an algorithm to be executed by the sensor node was defined (Algorithm 1). In short, the algorithm is based on sampling *k* times the sensors’ output, gathering them in one package, and transmiting the package after that. The predefined values of *k* were set to 1, 3, 5, 10, and 15 (*k* is limited to 15 due to the payload limitation of 251 bytes specified by the LoRa protocol). Five distinct configurations to sample and send data were considered to verify the best way to obtain more data during the time period.
**Algorithm 1**: Sensor Node Algorithm**Inputs**: *k***Output**: Transmission number of data packages1: {**Step 1**} Wake up the MCU and sensors and set n=02: **while**
n≤k**:**3:       {**Step 2**} Read the output of rain detector.4:       {**Step 3**} Read the output of the soil moisture sensor.5:       {**Step 4**} Read the outputs of a BME680 sensor.6:       {**Step 5**} Add the measured data in a LoRa package               and increments *n*.7:       {**Step 6**} Sleep for a predefined period of time.8: **end while**9: {**Step 7**} Wake up the MCU and radio and transmit the the LoRa             package.10: {**Step 8**} Sleep for a predefined period of time.11: {**Step 9**} Go back to **Step 1**.

The minimum output voltage that the LTC3108-1 can provide to the load is 2.5 V, and this voltage level was chosen to reduce power consumption. Regarding energy consumption of the sensor node, a measurement cycle consumes about 200 μJ and the load drains 8.4 μW during the sleep mode. Concerning transmission energy consumption, the payload length is a determinant factor. For the five configurations of measurement–transmission, the payload lengths are 21, 53, 84, 166, and 246 bytes and the consumed energies for transmitting each one are 1.15, 2.14, 3.13, 5.61, and 8.08 mJ, respectively. An estimation of the total number of transmission (*N*) for every configuration can be computed using Equation (Equation 8).
(8)$$N=\left\lfloor \frac{E_{T}-P_{sleep}\cdot \Delta t_{sleep}}{k\cdot E_{M}+E_{TX}}\right\rfloor$$
where $E_{T}$ is the available harvested energy, $P_{sleep}$ is the wasted power in sleep mode, $\Delta t_{sleep}$ is the total period in sleep mode, $k\cdot E_{M}$ is the consumed energy in the set of *k* measurement cycles, and $E_{TX}$ is the consumed energy to transmit the data. The total number of measurements is obtained by N×k. As the total available energy generated by the QC-based harvester was 1.38 J, the total number of transmissions, computed using the Equation (Equation 8), by each value of *k* is described in Table 1.

As we can observe in Table 1, it is possible to select the best configuration according to the application requirement, that is, there is a tradeoff between the maximum number of measurements and the maximum number of transmissions per day. For instance, the maximum number of measurements is 885, but a package is only transmitted every 24 min 24 s. On the other hand, a package transmission at every 2 min 58 s can be achieved, but only 484 measurements are taken.

## 6. Conclusions

The presented work describes a compact thermoelectric energy harvester based on a novel solar absorber and a water-cooled heat sink. The main novelty of the proposed energy harvester is the solar absorber based on a 100 μm thick quasicrystalline alloy as a coating for a copper block (40 × 40 × 5 mm size). The main characteristics of the solar absorber were described and explained with mathematical models. An evaluation testing was carried out to compare the performance of the proposed QC-based harvester with another similar energy harvester but with a solar absorber based on conventional black paint (BP). As a result, the QC-based harvester showed a better performance than the BP-based harvester in all the evaluated parameters. We also presented a study on how much the harvested energy can supply power to a Smart agriculture application (LoRa-based sensor node) during the day considering a tradeoff between the maximum number of taken measurements and the maximum number of transmissions per day. The results signaled that the proposed QC-based harvester is very suitable for powering this type of low-power application. In summary, the main contributions of this paper are described as follows:proposition of an original approach to generate energy using quasicrystal,development of a proof-of-concept for validating the potential of a quasicrystalline-based solar absorber to retain thermal energy,development of a complete energy harvesting system to generate electrical energy almost all day long, anda study estimating the potential to use the total energy provided by the proposed harvester to power a wireless sensor all day.

## Figures and Tables

**Figure 1 micromachines-12-00393-f001:**
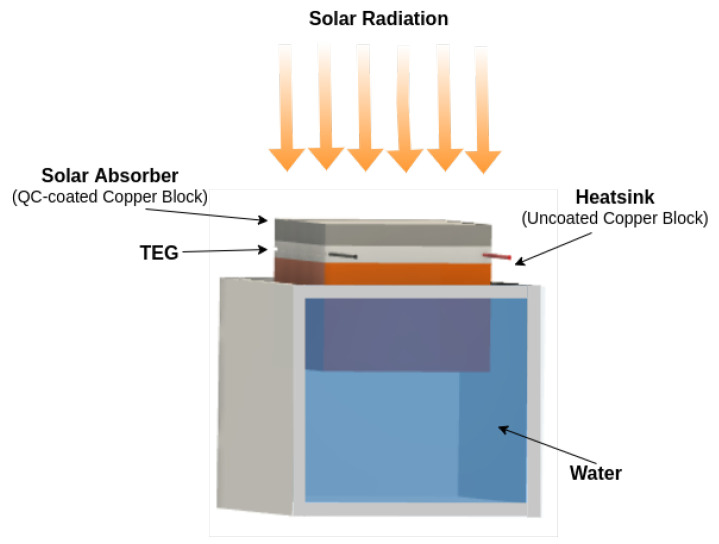
The proposed thermoelectric energy harvester.

**Figure 2 micromachines-12-00393-f002:**
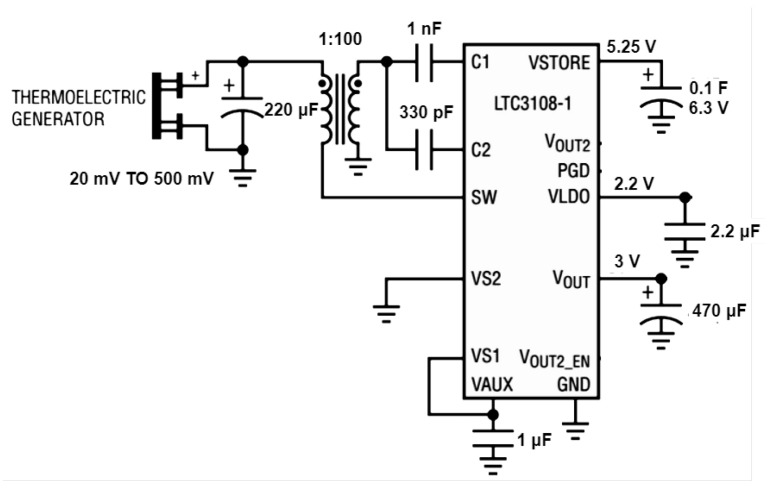
Typical schematic diagram of LTC3108-1 powered by a thermoelectric generator. Source: [30].

**Figure 3 micromachines-12-00393-f003:**
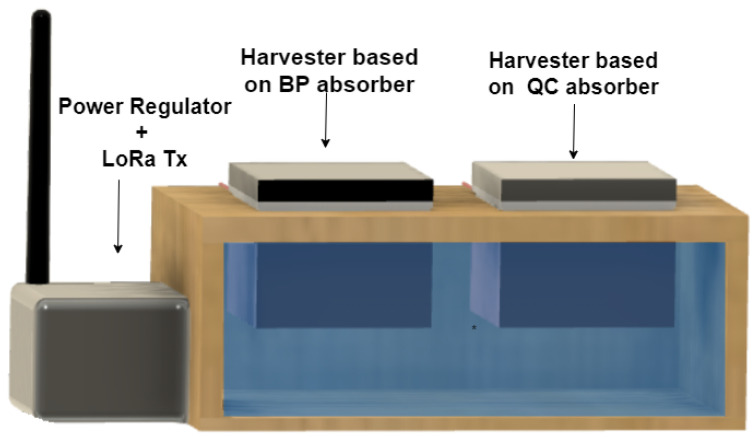
Experimental setup for testing.

**Figure 4 micromachines-12-00393-f004:**
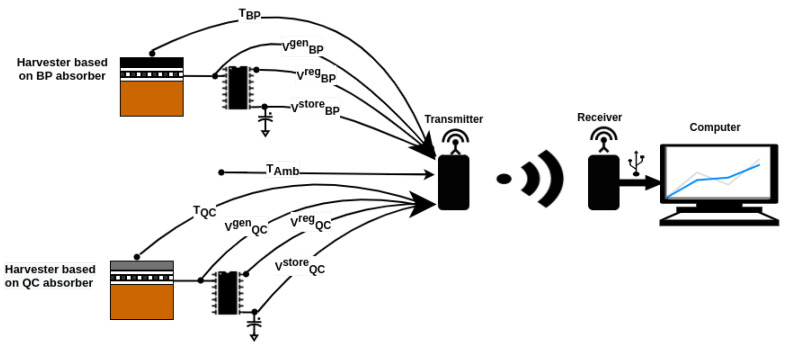
Schematic of the experimental setup of the thermoelectric energy harvester test.

**Figure 5 micromachines-12-00393-f005:**
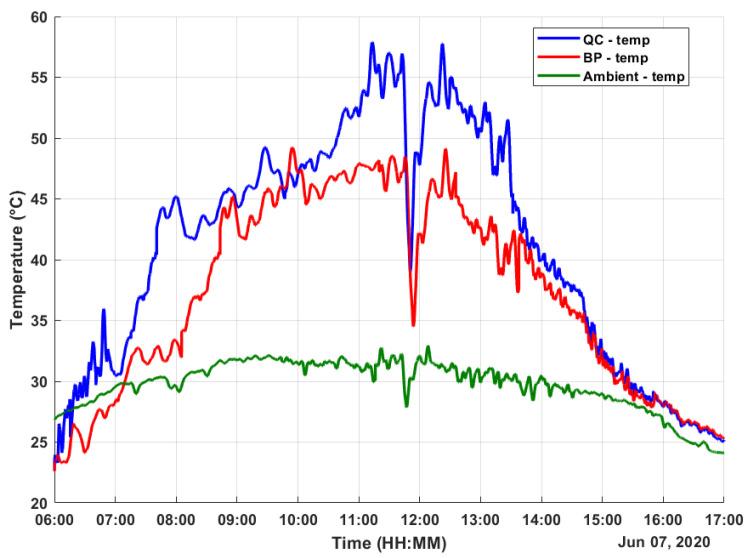
Obtained data: temperatures.

**Figure 6 micromachines-12-00393-f006:**
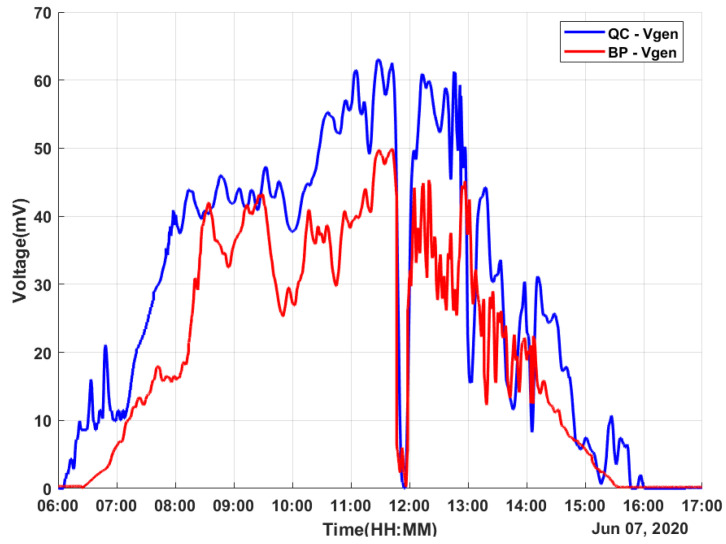
Obtained data: generated voltage.

**Figure 7 micromachines-12-00393-f007:**
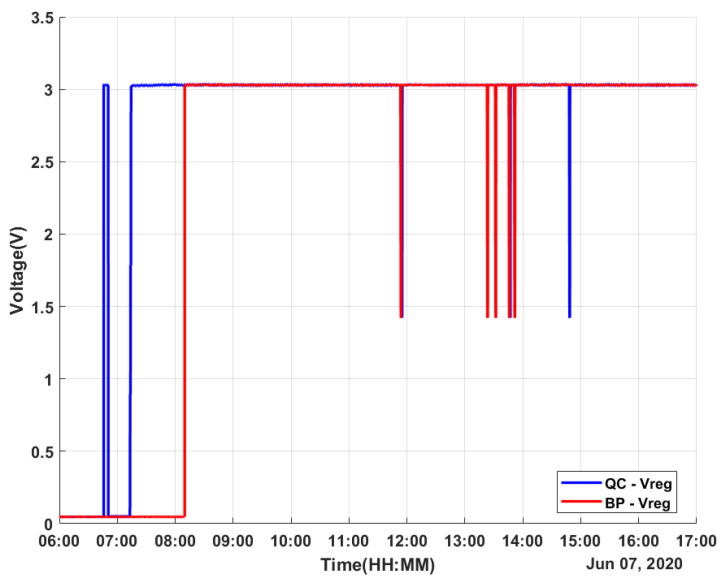
Obtained data: regulated voltage.

**Figure 8 micromachines-12-00393-f008:**
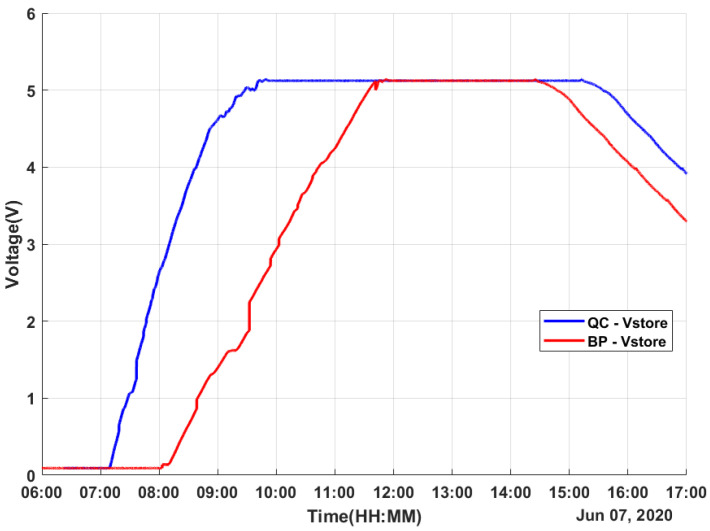
Obtained data: stored voltage.

**Figure 9 micromachines-12-00393-f009:**
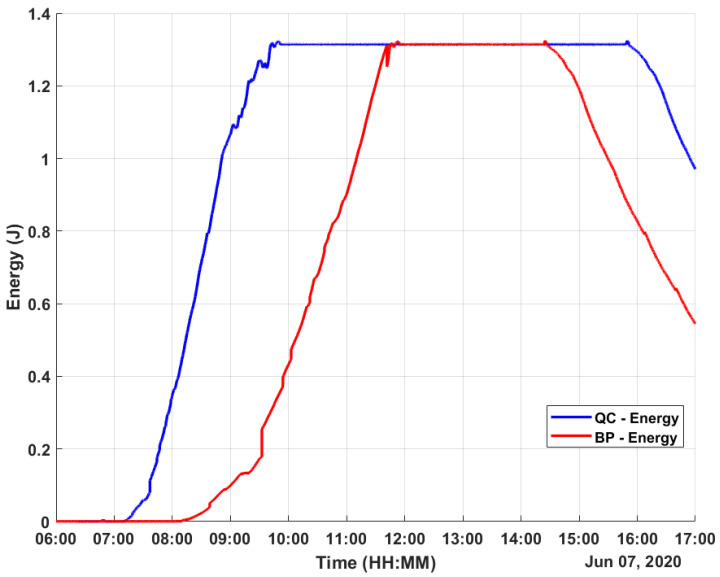
Obtained data: energy.

**Figure 10 micromachines-12-00393-f010:**
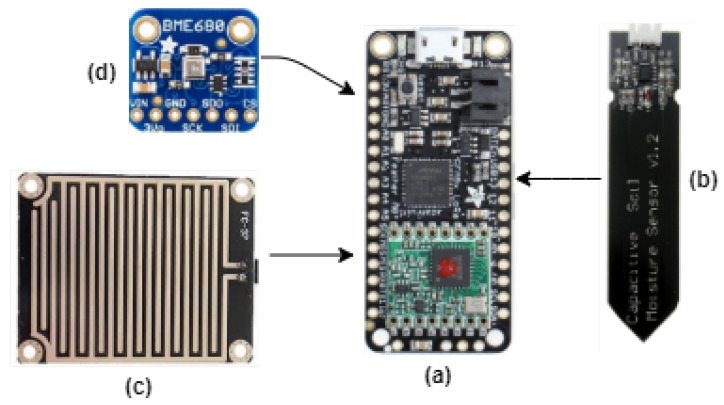
Sensor node. (**a**) Adafruit Feather M0+. (**b**) Capacitive soil moisture sensor. (**c**) Rain detector. (**d**) BME680 environmental sensor.

**Table 1 micromachines-12-00393-t001:** Sensor node algorithm results.

*k*	Measurement	Transmission
1	484 (every 2 min 58 s)	484 (every 2 min 58 s)
3	714 (every 2 min 1 s)	238 (every 6 min 3 s)
5	790 (every 1 min 49 s)	158 (every 9 min 7 s)
10	850 (every 1 min 42 s)	85 (every 16 min 57 s)
15	885 (every 1 min 38 s)	59 (every 24 min 24 s)

## Data Availability

Not Applicable.

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
