# Peer review of "A Thermoelectric Energy Harvester Based on Microstructured Quasicrystalline Solar Absorber"

_micromachines, 2021, doi:10.3390/mi12040393_

Round 1
Reviewer 1 Report
This work investigates a novel thermoelectric generator (TEG) using qusicrystal (QC) coating, which realizes a compact TEG-based energy harvester. Experimental results are just above average, and are not well presented. Verification results are not clearly illustrated. Discussion of references and literature about QC can be improved.
Below is a list indicating the major concerns:
- Line 131 to 134 states there are copper blocks and QC coated copper block in Fig. 1, but it is not clear where they are in Fig. 1.
- The T defined in Equations (1) and (2) is an average temperature, but the notation of T is also used in Equations (4) to (7), where they are not an average temperature – that is confusing.
- The discussion of QC in the Introduction is very similar as that in the previously published paper by the authors in [5].
- Through the Equations in the Section 2.1, I still cannot clearly capture the ideal how the QC coating helps to reduce emittance (epsilon) and enhance absorptance (alpha). Literature review on basic optical properties of QC should be incorporated in Section 2.1 and/or Introduction.
Also, it is noted the manufacturing parameters (such as composition, deposition technique, and heat treatment conditions) affect the materials properties of QC significantly (see below the literature, which is published by Prof. D. Shechtman who won the Nobel Prize in Chemistry owing to his discovery of QC), the results of this manuscript will be more convincing if the authors specify more details about their QC coatings and HVOF (Line 184 to 188), and/or cite more adequate references on QC deposition using HVOF.
Lang, C.I., Sordelet, D.J., Besser, M.F. et al. Quasicrystalline coatings: Thermal evolution of structure and properties. Journal of Materials Research 15, 1894–1904 (2000). https://doi.org/10.1557/JMR.2000.0275
- BP (black paint) is first appeared in Page 3 Line 124-125 and should be defined there, not in Page 7 Line 219-220.
- Result in Fig. 6 conflicts with result in Fig. 5: before 7:00 AM, the delta-T of the BP one is negative, how come there is generated voltage (6:00-7:00 AM, red curve in Fig. 6)? Please explain.
- Also, in Fig. 5 the delta-T after 15:00 PM is trivial, so that the generated voltage is null (Fig. 6) and stored voltage is declined (Fig. 8), however, the regulated voltage of both QC and BP set-ups are still 3.0 V (Fig. 7). It’s not clear how the ADC (development board) works with the LTC3108-1 module since the bkg voltage of ADC is 35 mV (Line 241) and the threshold of charging circuit is 20 mV (Line 203).
- Typo in Line 296-297: were developed was developed
- It’s suspected the time-axis of data in Fig. 5 and Fig. 6 has a shift or non-linear scaling between QC and BP curves before 11:30 AM. Specifically, in the period of 7:30 AM to 10:30 AM, there is an obvious time-axis shift between the QC and BP data. Temperature between QC and BP shouldn’t present a lag as large as 1 hour. Please check and explain.
- As for the verification in Section 5, the authors claimed they performed measurement for 24 hours (Line 324), but they used the 1.38 J harvested energy on June 07, 2020 (Line 342) to discuss about the results in Table 2. It is unclear the results in Table 2 are measured through a single experiment or estimated from two individual experiments. Also, the 24-hour sensor node measuring time is not suitable to the TEG because there is no solar irradiation in the night.
The above mentioned concerns have to be addressed before this manuscript is accepted for publication by the journal of Micromachines.
Reviewer 2 Report
There is no doubt that the QC based thermoelectric energy harvester has potential for wireless sensors application. Authors presented QC based TEG with water cooled heat sink. The commercial TEG tested with the power conditioning circuitry to understand the voltage and power level with the variation of temperature. The experimental results have been discussed and analysed with theory which is good. The manuscript is well written and formatted and the significant contribution made on this topic. I will recommend publishing this article in your reputed Journal however I would recommend to make the following mandatory corrections:
1.Authors claim the novelty of the proposed energy harvester using a solar absorber coated by a micrometric quasicrystalline alloy but this need to justify.
2.In their experiment , authors used commercial TEG: TEC1-12706 & power conditioning circuit LTC3108-1. Perhaps it would be better to discuss why they chosen these specific products.
Reviewer 3 Report
This paper presented a thermoelectric energy harvester based on micro structured quasi crystalline solar absorber. It should be improved with the following suggestions:
- Authors should briefly state the drawbacks in each of the papers reviewed
- Please provide a justification for using the type of solar absorber you used
- The main findings from the study should be presented with bullet points in the conclusion section for easy understanding
- Please provide some details about the cost implication of the proposed QC-based harvester compared to the similar harvester used in the study
- Please explain if the results and conclusions obtained from this study will be different in other climatic regions since the experimental system was deployed in Brazil
Round 2
Reviewer 1 Report
Thanks for the revisions.